# Towards Better Evaluation for
# Dynamic Link Prediction

**Farimah Poursafaei**[*]**, Shenyang Huang**[*]**, Kellin Pelrine, Reihaneh Rabbany**
McGill University School of Computer Science, Mila – Quebec AI Institute
[farimah.poursafaei,huangshe,kellin.pelrine,reihaneh.rabbany]@mila.quebec

## Abstract

Despite the prevalence of recent success in learning from static graphs, learning from time-evolving graphs remains an open challenge. In this work, we design new, more stringent evaluation procedures for link prediction specific to dynamic graphs, which reflect real-world considerations, to better compare the strengths and weaknesses of methods. First, we create two visualization techniques to understand the reoccurring patterns of edges over time and show that many edges reoccur at later time steps. Based on this observation, we propose a pure memorization-based baseline called EdgeBank. EdgeBank achieves surprisingly strong performance across multiple settings which highlights that the negative edges used in the current evaluation are easy. To sample more challenging negative edges, we introduce two novel negative sampling strategies that improve robustness and better match real-world applications. Lastly, we introduce six new dynamic graph datasets from a diverse set of domains missing from current benchmarks, providing new challenges and opportunities for future research. Our code repository is accessible at https://github.com/fpour/DGB.git.

## 1 Introduction

Many evolving real-world relations can be modelled by a dynamic graph where nodes correspond to entities and edges represent relations between nodes. Nodes, edges, weights or attributes in a dynamic graph can be added, deleted or adjusted over time. Therefore, understanding and analyzing the temporal patterns of a dynamic graph is an important problem. For instance, in popular online social networks, many users join the platform on a daily basis while connections between users are constantly added or removed [10]. To facilitate more efficient learning on dynamic graphs, many efforts have been devoted to the development of dynamic graph representation learning methods [40, 37, 41, 28, 42, 43, 29, 3].

Link prediction is a fundamental learning task on dynamic graphs which focuses on predicting future connections between nodes. Recent methods such as [18, 38, 42, 28, 41] show promising performance on this task, with the state-of-the-art (SOTA) performance [28, 41] being close to perfect on most existing benchmark datasets. However, considering that link prediction in static graphs, an arguably less complex task, still faces major challenges [12, 11], it is important to meticulously examine the near-perfect performance of dynamic link prediction methods. We hypothesize that current evaluation procedures and datasets fail to properly differentiate between the proposed approaches. Therefore, we identify several limitations in current evaluation procedures and propose solutions towards more robust and effective evaluation protocols.

**Limited Domain Diversity.** Existing benchmark datasets are mostly social or interaction networks thus limited in domain diversity. It is well-known that networks across different domains exhibit a

---

[*]Equal contribution.

36th Conference on Neural Information Processing Systems (NeurIPS 2022), Datasets and Benchmarks Track.

diverse set of properties. For example, biological networks such as protein interaction networks differ significantly from social networks in community structure and centrality measures [9]. Therefore, it is necessary to test dynamic link prediction methods in various domains outside of social or interaction networks. To this end, we incorporate six new datasets for dynamic link prediction ranging from politics, economics, and transportation networks. In addition, we introduce novel visualization techniques for dynamic graphs. We show that in most networks, a significant portion of edges reoccur over time but the reocurrence patterns vary widely across different networks and domains.

**Easy Negative Edges.** In a dynamic network, the edges that have been never observed during previous timestamps can be considered as *easy* negative edges, since it is less likely that these edges occur during the test phase given the reoccurring pattern of dynamic graphs. We introduce two novel Negative Sampling (NS) strategies designed specifically for dynamic graphs which consider more difficult negative edges based on the reoccurrence pattern of observed edges. As shown in Fig. 1, SOTA methods have a significant decrease in performance when a different set of negative edges are used for evaluation. Moreover, the relative ranking of methods varies significantly across NS settings. Therefore, it is important to evaluate against different sets of negative edges to better understand the performance of different models.

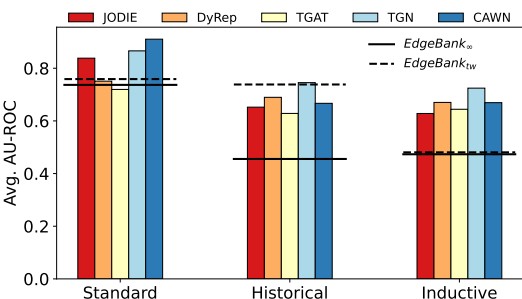

Figure 1: The ranking of different methods changes in the proposed negative sampling settings which contains more difficult negative edges. Our proposed baselines (horizontal lines) show competitive performance, in particular in the standard setup. The results illustrate the average performance over all datasets presented in Table 1.

**Memorization Works Well.** Finally, we introduce a simple memorization-based baseline, named EdgeBank, which simply stores previously observed edges in memory, and then predicts stored edges in memory as positive at test time. In Fig. 1, we contrast the performance of SOTA methods with that of EdgeBank (in horizontal lines). EdgeBank is a surprisingly strong baseline for dynamic link prediction. In the historical NS setting, EdgeBank achieves the second best ranking amongst all methods. As EdgeBank requires neither learning nor hyper-parameter tuning, we argue that it is a strong and necessary baseline for future methods to compare against.

The goal of this work is to propose more effective evaluation strategies to better differentiate dynamic link prediction methods. We identify challenges and drawbacks in the current evaluation setting for dynamic link prediction: (1) existing strategies for sampling negative edges during evaluation are insufficient, (2) memorization leads to over-optimistic evaluation, and (3) there is a lack of diversity in dynamic graph dataset domains. Our main contributions can be summarized as follows:

- **Novel Negative Sampling Strategies**. We evaluate the impact of negative edges on model performance and outline two novel sampling strategies: *historical NS* and *inductive NS*, which provide more robust and in depth evaluation.
- **Strong Baseline**. We propose a novel non-parameterized and memorization-based method, EdgeBank, which provides a strong baseline for current and future approaches to compare against.
- **New Datasets and Visualization Tools**. We present six novel dynamic graph datasets from various domains such as politics, transportation, and economics. These datasets exhibit different temporal edge evolution patterns, which can be understood through our proposed TEA and TET plots.

**Reproducibility**: our code repository is available at https://github.com/fpour/DGB.git. All datasets can be accessed at https://zenodo.org/record/7008205#.Yv_a_3bMJPZ.

## 2 Related Work

**Benchmarking Graph Learning Methods.** A number of studies identify several issues in evaluation of existing GNN models [5, 33, 6, 12, 22]. Focusing on static graphs, Dwivedi et al. [5] identify issues with comparative evaluation due to inconsistent experimental settings. Shchur et al. [33] show that reusing the same train-test splits in many different works has led to overfitting and using different splits of the data could result in different ranking of the methods. OGB [12] facilitates reproducibility and scalability of graph learning tasks by providing a diverse set of datasets together with unified

evaluation protocols, metrics, and data splits. In contrast to these works, we focus on improving evaluation for *dynamic* link prediction.

For dynamic graphs, Junuthula et al. [14] differentiate dynamic and static link prediction by edge insertion or deletion. Junuthula et al. [15] then consider the problem of incorporating information from friendship networks into predicting future links in social interaction domains. Haghani and Keyvanpour [10] provide a comprehensive review of link prediction methods for social networks and categorize the link prediction task into two groups: missing link prediction, and future link prediction. Similar to these works, we also focus on dynamic link prediction but from different perspectives: new negative sampling strategies, new baseline, and new dataset domains.

**Negative Sampling (NS) of Edges in Graphs.** Yang et al. [45] argue that NS is as important as positive sampling in graph representation learning. For static link prediction, the most common method is to sample negative edges at random [8, 1, 32]. Alternatively, the sampling can be based on connecting nodes with specific properties (e.g. a sufficiently large degree) [19], or it can be based on a particular geodesic distance [20, 21]. Kotnis and Nastase [17] provide an empirical study of the impact of different NS strategies during training on the learned representations of various methods in knowledge graphs. In our work, we focus on the impact of NS strategies during evaluation, and propose two novel NS strategies based on the history of the observed edges in dynamic graphs. Current evaluation protocol has difficulty differentiating between models as many methods achieve near-perfect performance across the board. In comparison, our proposed NS strategies sample harder negative edges for better evaluation.

**Dynamic Graph Representation Learning.** Recently there is a surge of interest towards temporal networks. Kazemi et al. [16] present a survey of advances in representation learning on dynamic graphs. Skardinga et al. [35] concentrate on recent studies on Dynamic Graph Neural Networks (DGNNs) and provide a detailed terminology of dynamic networks. Zhang et al. [46] highlights the importance of learning fully temporal embeddings which also models information propagation. Skardinga et al. [35] and Kazemi et al. [16] both argue modeling dynamic graphs with continuous representations has higher potential, since it offers superior temporal granularity. In our experiments we center our attention on five recent models of this type: *JODIE* [18], *DyRep* [38], *TGAT* [42], *TGN* [28], and *CAWN* [41]. An overview of these methods is provided in Appendix A.1. As shown in Section 6, these methods often achieve close to perfect performance for current link prediction tasks on dynamic graphs. This hinders researchers' ability to evaluate if new models are superior. Also, it exaggerates the efficacy of current models on real-world tasks. Hence, we further examine the evaluation procedure, from the perspective of benchmark datasets, negative sampling and baselines.

# 3 Understanding Dynamic Graph Datasets

A dynamic graph can be represented as timestamped edge streams – triplets of source, destination, timestamp, i.e. $\mathcal{G} = \{(s_0, d_0, t_0), (s_1, d_1, t_1), \ldots, (s_T, d_T, T)\}$ where the timestamps are ordered ($0 \leq t_1 \leq t_2 \leq \ldots \leq t_{split} \leq \ldots \leq T$). We investigate the task of predicting the existence of an edge between a node pair in the future. The timeline is split at a point, $t_{split}$, into all edges appearing before or after. This results in train and test edge sets $E_{train}$ and $E_{test}$. We can then divide edges of a given dynamic graphs into three categories: (a) $E_{train} \setminus E_{test}$: edges that are only seen during training, (b) $E_{train} \cap E_{test}$: edges that are seen during training and reappear during test, which can be considered as *transductive* edges, and (c) $E_{test} \setminus E_{train}$: edges that have not been seen during training and only appear during test, which can be considered as *inductive* edges.

We aim to understand the differences between dynamic graph datasets across a variety of domains. To this end, we first investigate seven widely used benchmark datasets and contribute six novel dynamic graphs (marked as *new*) from diverse domains currently under-studied in dynamic link prediction literature. The statistics of these datasets are summarized in Table 1, and details are explained in Section 3.1. To better characterize the differences between dynamic graphs, we propose two types of plots and define three indices to visualize and quantify the patterns in dynamic graphs and the difficulty of a given evaluation split in Section 3.2 and Section 3.3.

Table 1: Dataset statistics.

| Dataset | Domain | # Nodes | Total Edges | Unique Edges | Unique Steps | Time Granularity | Duration |
|---------|--------|---------|-------------|--------------|--------------|------------------|----------|
| Wikipedia | Social | 9,227 | 157,474 | 18,257 | 152,757 | Unix timestamp | 1 month |
| Reddit | Social | 10,984 | 672,447 | 78,516 | 669,065 | Unix timestamp | 1 month |
| MOOC | Interaction | 7,144 | 411,749 | 178,443 | 345,600 | Unix timestamp | 17 month |
| LastFM | Interaction | 1,980 | 1,293,103 | 154,993 | 1,283,614 | Unix timestamp | 1 month |
| Enron | Social | 184 | 125,235 | 3,125 | 22,632 | Unix timestamp | 3 years |
| Social Evo. | Proximity | 74 | 2,099,519 | 4,486 | 565,932 | Unix timestamp | 8 months |
| UCI | Social | 1,899 | 59,835 | 20,296 | 58,911 | Unix timestamp | 196 days |
| Flights (new) | Transport | 13,169 | 1,927,145 | 395,072 | 122 | days | 4 months |
| Can. Parl. (new) | Politics | 734 | 74,478 | 51,331 | 14 | years | 14 years |
| US Legis. (new) | Politics | 225 | 60,396 | 26,423 | 12 | congresses | 12 congresses |
| UN Trade (new) | Economics | 255 | 507,497 | 36,182 | 32 | years | 32 years |
| UN Vote (new) | Politics | 201 | 1,035,742 | 31,516 | 72 | years | 72 years |
| Contact (new) | Proximity | 694 | 2,426,280 | 79,531 | 8,065 | 5 minutes | 1 month |

## 3.1 Temporal Graph Datasets

We consider a wide set of dynamic graph datasets from diverse domains. The data collection and processing details are explained in Appendix A.2. All datasets are publicly available under MIT licence or Apache License 2.0. Note that none of these datasets contains node attributes, but we include description of edge attributes when applicable.

- **Wikipedia** [18]: consists of edits on Wikipedia pages over one month. Editors and Wiki pages are modelled as nodes, and the timestamped posting requests are edges. Edge features are LIWC-feature vectors [27] of edit texts with a length of 172.
- **Reddit** [18]: models subreddits' posted spanning one month, where the nodes are users or posts and the edges are the timestamped posting requests. Edge features are LIWC-feature vectors [27] of edit texts with a length of 172.
- **MOOC** [18]: is a student interaction network formed from online course content units such as problem sets and videos. Each edge is a student accessing a content unit and has 4 features.
- **LastFM** [18]: is an interaction network where users and songs are nodes and each edge represents a user-listens-to-song relation. The dataset consists of the relations of 1000 users listening to the 1000 most listened songs over a period of one month. The dataset contains no attributes.
- **Enron** [34]: is an email correspondence dataset containing around 50K emails exchanged among employees of the ENRON energy company over a three-year period. This dataset has no attributes.
- **Social Evo.** [24]: is a mobile phone proximity network which tracks the everyday life of a whole undergraduate dormitory from October 2008 to May 2009. Each edge has 2 features.
- **UCI** [26]: is a Facebook-like, unattributed online communication network among students of the University of California at Irvine, along with timestamps with the temporal granularity of seconds.
- **Flights** (*new*) [31]: is a directed dynamic flight network illustrating the development of the air traffic during the COVID-19 pandemic. It was extracted and cleaned for the purpose of this study. Each node represents an airport and each edge is a tracked flight. The edge weights specify the number of flights between two given airports in a day.
- **Can. Parl.** (*new*) [13]: is a dynamic political network documenting the interactions between Canadian Members of Parliaments (MPs) from 2006 to 2019. Each node is one MP representing an electoral district and each edge is formed when two MPs both voted "yes" on a bill. The edge weights specify the number of times that one MP voted "yes" for another MP in a year.
- **US Legis.** (*new*) [7, 13]: is a senate co-sponsorship graph which documents social interactions between legislators from the US Senate. The edge weights specify the number of times two congress persons have co-sponsored a bill in a given congress.
- **UN Trade** (*new*) [23]: is a weighted, directed, food and agriculture trading graph between 181 nations and spanning over 30 years. The edge weights specify the total sum of normalized agriculture import or export values between two countries.
- **UN Vote** (*new*) [39]: is a dataset of roll-call votes in the United Nations General Assembly from 1946 to 2020. If two nations both voted "yes" for an item, then the edge weight between them is incremented by one.
- **Contact** (*new*) [30]: is a dataset describing the temporal evolution of the physical proximity around 700 university students over a period of four weeks. Each participant is assigned an unique ID and edges between users indicate that they are within close proximity of each other. The edge weights indicate the physical proximity between participants.

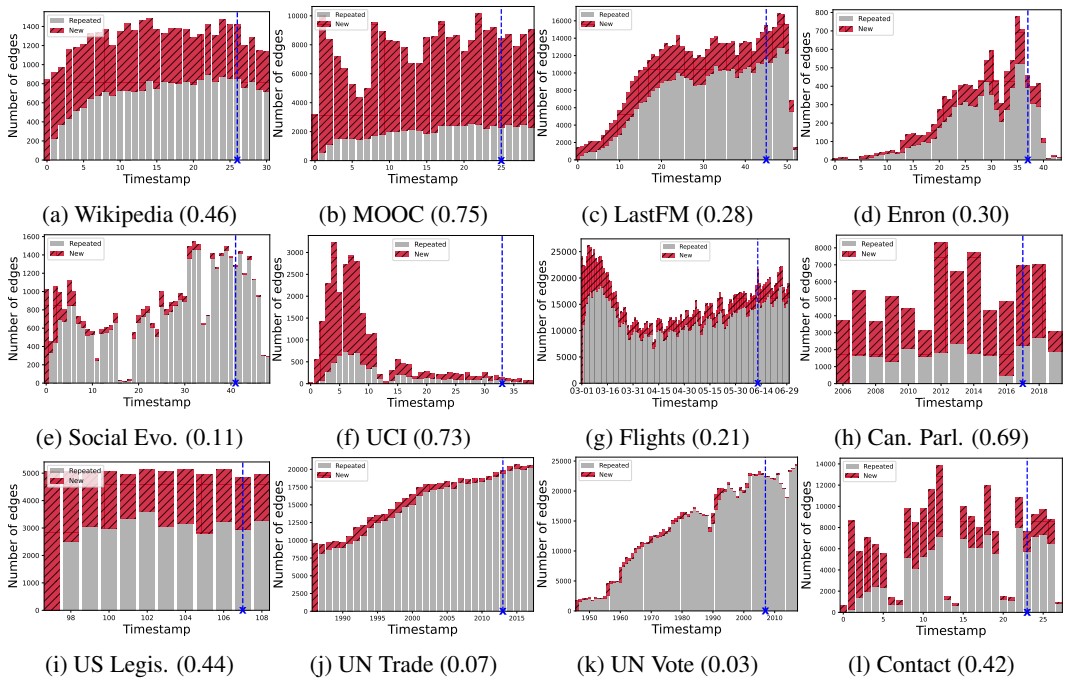

(a) Wikipedia (0.46)  (b) MOOC (0.75)  (c) LastFM (0.28)  (d) Enron (0.30)

(e) Social Evo. (0.11)  (f) UCI (0.73)  (g) Flights (0.21)  (h) Can. Parl. (0.69)

(i) US Legis. (0.44)  (j) UN Trade (0.07)  (k) UN Vote (0.03)  (l) Contact (0.42)

Figure 2: TEA plots show many real-world dynamic networks contain a large proportion of edges that reoccur over time. Thus, even a simple memorization approach such as EdgeBank can potentially achieve strong performance. The numbers in parentheses report the novelty index. Due to space limitation, the Reddit's TEA plot is presented in Fig. 7a in Appendix A.3.

## 3.2 Temporal Edge Appearance (TEA) Plot

A TEA plot illustrates the portion of repeated edges versus newly observed edges for each timestamp in a dynamic graph, as shown in Fig. 2. The grey bar indicates the number of edges which were observed in previous time steps and the red bar represents the number of new edges seen at each step. To further quantify the observed pattern, we measure the average ratio of new edges in each timestamp as:

$$
novelty = \frac{1}{T} \sum_{t=1}^{T} \frac{|E^t \setminus E^t_{seen}|}{|E^t|}, \text{ where } E^t = \{(s, d, t_e)| \, t_e = t\} \text{ and } E^t_{seen} = \{(s, d, t_e)| \, t_e < t\}
$$

Here, $E^t$ denotes the set of edges present in timestamp $t$, and $E^t_{seen}$ denotes the set of all edges seen in the previous timestamps. This metric gives an estimation of the portion of positive edges that a pure memorization method cannot predict correctly.

Fig. 2 shows high variance across datasets in temporal evolutionary patterns in terms of new and repeated edges. Some datasets such as Social Evo. comprise mainly repeated edges, while others such as MOOC have a high proportion of new edges. The TEA plots also show significant differences in when edges occur, and distinctions between our new datasets and existing ones. For example, our new Flights dataset has significantly more unique edges and higher numbers of edges per timestamp.

TEA plots show that it is important to consider the relative distribution of the repeated and new edges when designing and choosing methods for the dynamic link prediction task. When many edges are repeated, a simple memorization approach can potentially achieve strong performance. In contrast, if there are many new edges, memorization would be insufficient. In addition, to understand the consistency of edge reoccurrence patterns, we now propose:

## 3.3 Temporal Edge Traffic (TET) Plot

A TET plot visualizes the reoccurrence pattern of edges in different dynamic networks over time, as shown in Fig. 3. To construct these plots, we first sort edges based on the timestamp they first appeared. Then for edges occurring in the same timestamp, we sort them based on when they last

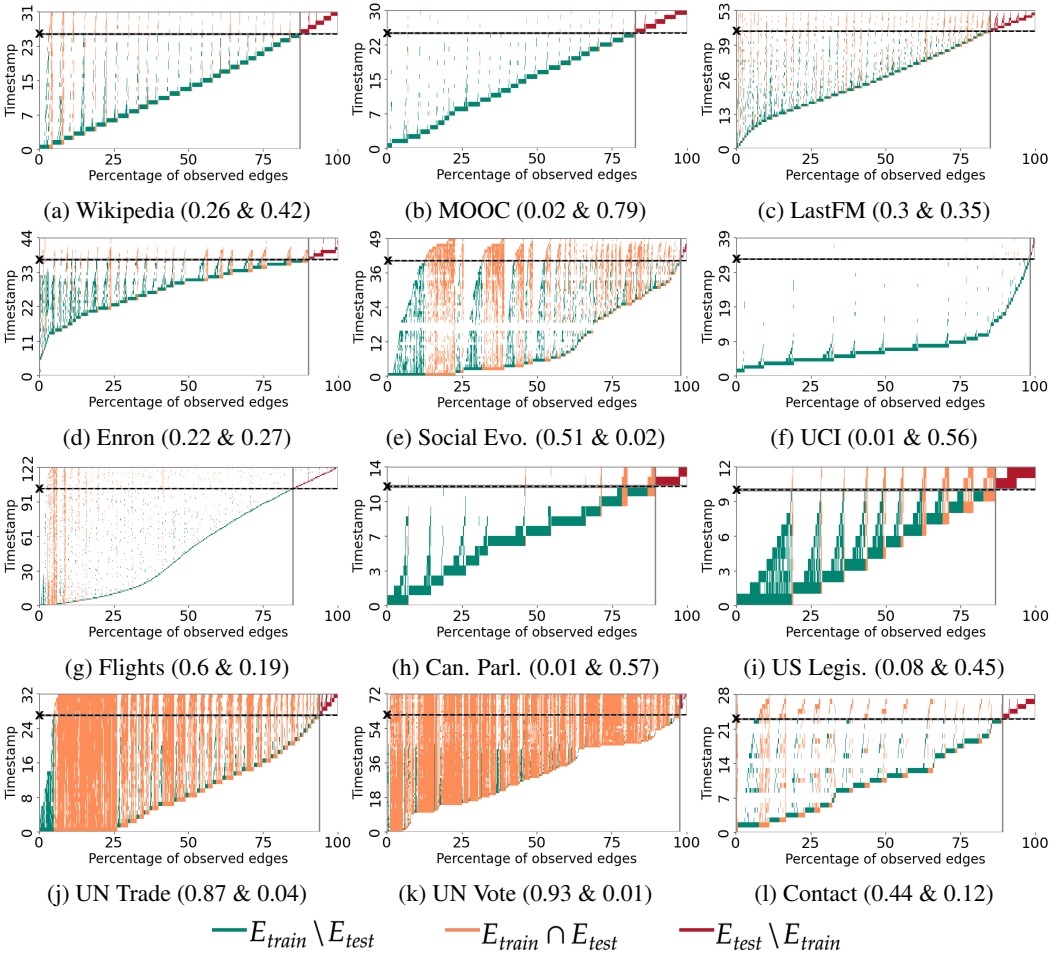

Figure 3: TET plots illustrate varied edge traffic patterns in different temporal graphs. The horizontal line starting with "**x**" marks $t_{split}$. In parentheses, we report the proportion of training set edges reoccurring in the test set (*reocurrence* index) & the proportion of unseen test edges (*surprise* index), respectively. Due to space limitation, the Reddit's TET plot is presented in Fig. 7b in Appendix A.3.

occurred. Further, we color edges based on whether they are seen in training set only (green), test set only (inductive edges, red), or both (transductive edges, orange). To quantify the patterns in these plots we define the following two indices:

$$reocurrence = \frac{|E_{\text{train}} \cap E_{\text{test}}|}{|E_{\text{train}}|} , \quad surprise = \frac{|E_{\text{test}} \setminus E_{\text{train}}|}{|E_{\text{test}}|}$$

TET plots provide more insights about the edges that are used for training and testing of different DGNN methods. A memorization approach can potentially predict the transductive positive edges, since it has observed and hence recorded them during training. In particular, if they appear consistently, then simple memorization is likely to be successful. This is when reocurrence index is high and surprise index is low. On the other hand, if they appear at some times but then disappear later, then memory is likely still helpful, but simple and full memorization will not work. It would incorrectly predict that those edges still exist, i.e. when reocurrence index is low. Meanwhile, memorization is not helpful at all for predicting inductive positive test edges at their first appearance, since these are new edges that have not been observed before, i.e. high surprise index.

We encourage researchers to investigate the proposed TEA and TET plots to get a more comprehensive overview of dynamic graphs in addition to the network statistics. For example, while Social Evo. and UN Trade have a relatively similar proportion of repeated vs. new edges based on their TEA plots, we see in their TET plots that UN Trade has far more consistent reocurrence. The clear difference we can observe in the visualization is mirrored in the results - the best model on UN Trade is among the worst on Social Evo., and vice versa (see Fig. 5).

# 4 EdgeBank: A Baseline for Dynamic Link Prediction

We propose a pure memorization-based approach called EdgeBank, in order to understand whether memorizing past edges can be a competitive baseline. This is based on the observation that many edges in dynamic graphs reoccur over time. The memory component of EdgeBank is simply a dictionary which is updated with newly observed edges at each timestamp, similar to the memory update procedure of TGN [28]. In this way, EdgeBank resembles a *bank* of observed edges and requires no parameters. The storage requirement of EdgeBank is the same as the number of edges in the dataset.

At test time, EdgeBank predicts a test edge as *positive* if the edge was seen before (stored in the memory), and *negative* otherwise. EdgeBank can accurately predict edges which reoccur frequently over time. There are two types of edges of which EdgeBank will make an incorrect prediction: (i) an unseen (inductive) edge, or (ii) an edge observed before (in memory) but are not observed at the current time. In the standard random negative sampling evaluation [28, 42, 41], as graphs are often sparse, it is unlikely that an edge observed before will be sampled as a negative edge. Therefore, EdgeBank has strong performance on negative edges in many cases.

We consider two different memory update strategies for EdgeBank thus resulting in two variants:

- **EdgeBank$_\infty$** stores all observed edges in memory, thus remembering edges even from a long time ago. It is prone to false positives on edges which appear once but rarely reoccur over time.
- **EdgeBank$_{tw}$** only remembers edges from a fixed size time window from the immediate past. The size of the time window is set to the duration of validation split, based on the intuition of predicting the test set behavior from the most similar (recent) period available. Hence, EdgeBank$_{tw}$ focuses on the edges observed in the short-term past.

Note that EdgeBank is not designed to replace state-of-the-art methods. Rather we argue that all dynamic graph representation methods should be able to do better than memorization, thus outperforming EdgeBank. EdgeBank provides a simple and strong baseline to demonstrate how far pure memorization can go on each dataset.

# 5 Revisiting Negative Sampling in Dynamic Graphs

Current SOTA methods for dynamic link prediction often achieve near perfect performance on existing benchmark datasets [18, 38, 42, 28, 41, 37]. Consequently, one can argue that either the existing datasets are too simplistic or the current evaluation process is insufficient to differentiate methods. We discussed the dataset aspect extensively. Next, we need to carefully examine the current evaluation setting of DGNNs. In particular, although negative edges constitute half of the evaluation edges, little attention has been dedicated to understanding the effect of different sets of negative edges on the overall performance. In this section, we take a closer look at Negative Sampling (NS) strategies for evaluation of dynamic link prediction, and propose two novel NS strategies for more robust evaluation and better differentiation amongst methods. To better motivate the two new methods, we first explain the standard random NS strategy widely used in literature.

**Random Negative Sampling.** Current evaluation samples negative edges randomly from almost all possible node pairs of the graphs [18, 38, 42, 28, 41]. At each time step, we have a set of positive edges consisting of source and destination nodes together with edge timestamps and edge features. To generate negative samples, the standard procedure is to keep the timestamps, features, and source nodes of the positive edges, while choosing destination nodes randomly from all nodes. This approach has two significant issues:

**(1) No Collision Checking**: most existing implementations have no collision check

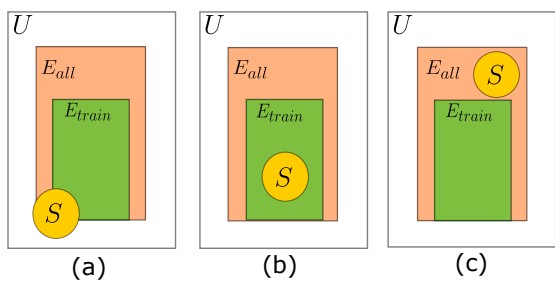

Figure 4: Negative edge sampling strategies during evaluation for dynamic link prediction; (a) random sampling (standard in existing works), (b) historical sampling (ours), (c) inductive sampling (ours).

between positive and negative edges. There are some exceptions, such as [3], but this holds for all the DGNN methods examined in our experiments. Therefore, it is possible for a given edge to be both positive and negative. This collision is more likely to happen in denser datasets, such as UN Vote and UN Trade. A basic accept-reject sampling could address this issue, as applied in our experiments.

**(2) No Reoccurring Edges**: the probability of sampling an edge which was observed before is often very low due to the sparsity of the graph. Therefore, a simple method like EdgeBank can perform well on negative edges. However, in many real-world tasks such as flight prediction, correct prediction of the same edge for different time steps is particularly important. For example, predicting that there will be no flight between the north and south poles this week is not nearly as practical as predicting whether a standard, reoccurring commuter flight will be canceled.

To address the second issue, we need to sample from previously observed edges, which can be from the training or test set. This constitutes the two alternative NS strategies proposed here, illustrated in Fig. 4. Here, $S$ is the sample space for negative edges. Let $U$, $E_{all}$, $E_{train}$ be the set of all possible node pairs, all edges in the dataset (train and test) and all edges in the train set, respectively. Note that $E_{all} = E_{train} + E_{test}$ where $E_{test}$ is all edges in the test set. Lastly, we set $U_{neg} = U - E_{all}$. In random NS, we sample from edges $e \in U$, with the proportion from $E_{all}$ and $E_{train}$ regulated only by the sizes of those sets relative to U. To resolve the issues with random NS, in the following sections we propose *historical NS* and *inductive NS*.

**Historical Negative Sampling.** In historical NS, we focus on sampling negative edges from the set of edges that have been observed during previous timestamps but are absent in the current step. The objective of this strategy is to evaluate whether a given method is able to predict in which timestamps an edge would reoccur, rather than, for example, naively predicting it always reoccurs whenever it has been seen once. Therefore, in historical NS, for a given time step $t$, we sample from the edges $e \in (E_{train} \cap \overline{E_t})$.

**Inductive Negative Sampling.** While in historical NS we focus on observed training edges, in inductive NS, our focus is to evaluate whether a given method can model the reoccurrence pattern of edges only seen during test time. At test time, after observing the edges that were not seen during training, the model is asked to predict if such edges exist in future steps of the test phase. Therefore, in the inductive NS, we sample from the edges $e \in (E_{test} \cap \overline{E_{train}} \cap \overline{E_t})$ at time step $t$. As these edges are not observed during training, they are considered as *inductive* edges. Note that in both *historical* and *inductive* NS if the number of available negative edge of the given type is less than the number of positive edges, the remaining negative edges are sampled by the random NS strategy. See discussion in Appendix B.2.

## 6   Experiments

In this section, we present a comprehensive evaluation of the dynamic link prediction task on all 13 datasets with 5 SOTA methods. Our experimental setup closely follows [18, 38, 42, 28, 41]. The objective of the link prediction task is to predict the existence of an edge between a node pair at a given time. For all DGNN methods, we use a Multilayer Perceptron as the output layer for edge prediction, where concatenated node embeddings are inputs and the probability of the edge is the output. For all experiments, we use the same $70\% - 15\% - 15\%$ chronological splits for the train-validation-test sets as [42, 28, 41]. The average results over five runs are reported. The *Area Under Receiver Operating Characteristic (AU-ROC)* metric is selected as the main performance metric. We visualize the results for easier interpretation, but the exact numbers that produce the visualizations – and the equivalents with *Average Precision (AP)* – are presented in the Appendix B.1.

Fig. 5a compares the performance of all models under the standard random NS strategy. First, we observe significant variation in performance for all models across datasets. This supports the benefits of evaluation on datasets from different domains. Second, we observe a strong inconsistency in relative ranking amongst methods across datasets. For example, while CAWN achieves SOTA on most datasets, on MOOC and Social Evo. it performs significantly worse than several other models. Lastly, note that EdgeBank demonstrates competitive performance even when compared to SOTA methods. Despite its simplicity, EdgeBank outperforms highly parametrized and complex models on datasets such as LastFM, Enron and UN Trade.

Next, we examine the impact of NS strategies on performance. Fig. 5b and Fig. 5c shows the performance of different methods with the *historical NS* and *inductive NS* strategies, respectively.

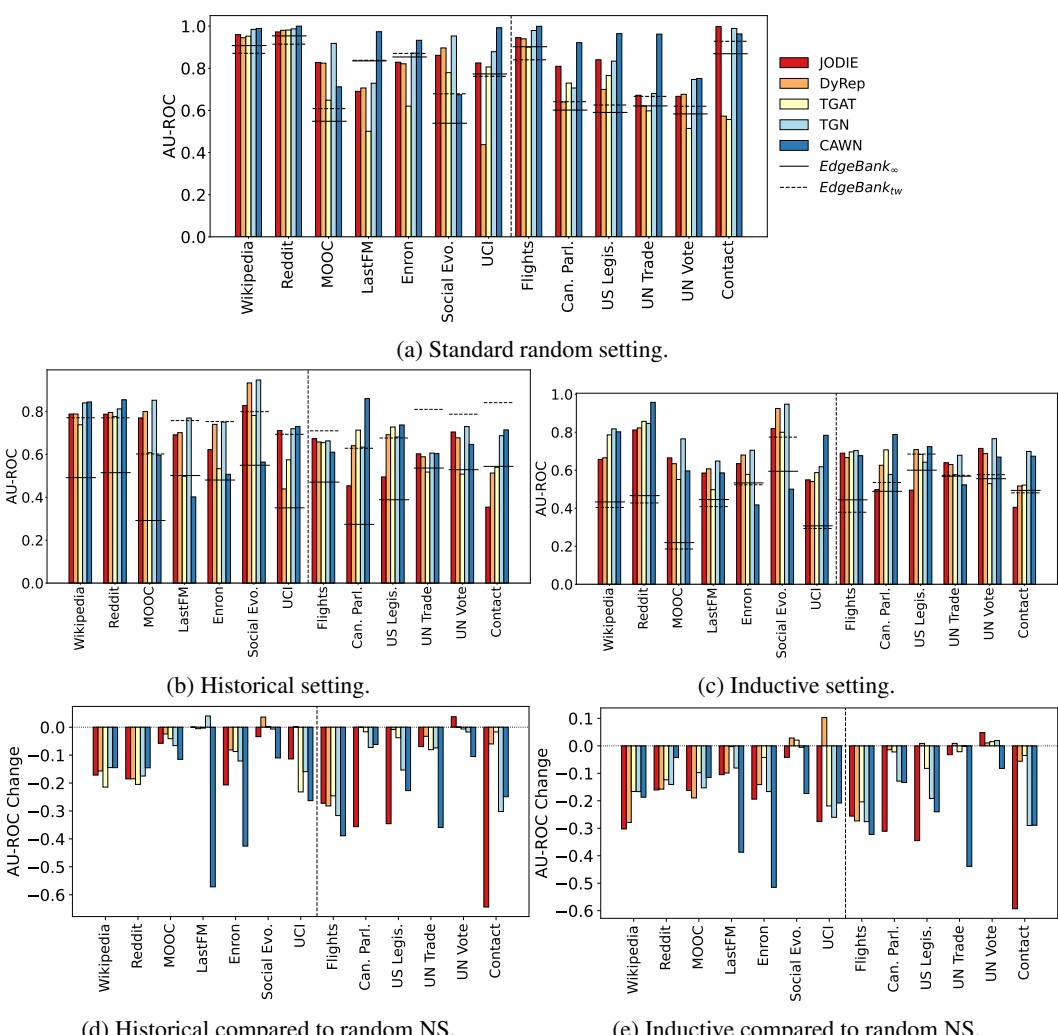

Figure 5: Performance of methods in all three NS settings. In (a) the proposed memorization baselines are on par with SOTA methods, and over-performing in some datasets, e.g. LastFM. In (b) and (c), with alternative negative sampling strategies, we observe a more clear gap between the performance of models and the memorization baseline, whilst the ranking of the models also changes, e.g. CAWN not being the ranked one in most datasets, which is in contrast with the rankings obtained in the standard setting. In (d) and (e), we report the performance drop when moving from the standard setting, which can hint at the (lack of) generalization power of different methods, especially in (e).

First, we observe that the ranking of models can change significantly across different NS settings. This shows that relying on a single NS strategy, such as the random NS, is insufficient for the complete evaluation of methods. Second, for the historical NS setting, EdgeBank$_{tw}$ becomes highly competitive, often beating most methods and even achieving SOTA for UN Trade, UN Vote, Flights, Enron, and Contact. This shows that in these datasets, recently observed edges contain crucial information for link prediction. Third, EdgeBank$_{\infty}$ has a significant drop in performance in both NS strategies. This shows that as the negative edges are sampled from either previously observed edges or unseen edges, naively memorizing all past edges is no longer sufficient. However, EdgeBank can perform competitively under *random* NS. This further shows that the standard *random* NS is limited in its ability to effectively differentiate methods. In Fig. 5d and Fig. 5e, we examine the performance changes for each model in historical or inductive NS setting. CAWN, which performed best overall with *random* NS, collapses on certain datasets such as LastFM and Enron. Other models fare much better on these datasets. All models exhibit a large performance drop on the Flights dataset.

The performance degradation is also correlated with the degree of memorization. Fig. 6 shows that the models which are more correlated with EdgeBank$_{\infty}$ tend to perform worse in the historical and inductive NS settings. Since EdgeBank$_{\infty}$ is naively dependent on the memory, higher correlation

with it indicates a model relies more heavily on memorization. For example, CAWN has the highest correlation and JODIE the second highest. They have the largest and second largest losses (respectively) in performance with the more challenging negative sampling. Similarly, DyRep is the least correlated with EdgeBank, and experiences the least drop in performance with historical NS and second least with inductive NS.

## 7   Conclusion

In this paper, we proposed tools to improve the evaluation of dynamic link prediction. First, we introduced *six new datasets* to increase the diversity of domains in which link prediction methods are currently being evaluated. Then, we created TEA and TET plots to visualize and quantify the temporal *patterns* of edges in dynamic graphs, and the difficulty of an evaluation split. Next, we showed the limitations of the current random negative sampling strategy used in the evaluation and introduced two new strategies, *historical* and *inductive* sampling, to better test the generalization of different models. Finally, we proposed a competitive yet simple memorization-based *baseline*, EdgeBank. It can yield insights into how much different models

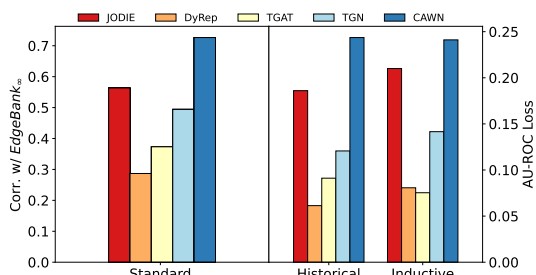

Figure 6: Performance correlation with the proposed memorization baseline, $EdgeBank_\infty$ (on the left), predicts the performance loss (lower = better) of the methods in both of the harder negative sampling settings (on the right).

rely on memorization. When we applied these tools to compare existing models, we found that the performance and ranking of different models vary significantly. We hope that these tools will lead to more thorough, lucid, and robust evaluation practices in dynamic link prediction.

**Broader Impact:** We expect this work to have a major impact on the fundamental as well as applied dynamic graph research. Essentially, high-quality datasets from diverse domains play undeniable roles in advancement of research (e.g., OGB [12] or ImageNet [4]). By contributing 6 new datasets from less explored real-world domains, we aim to enrich available datasets for dynamic graph learning tasks, and facilitate the development of novel dynamic graph models. In addition, our proposed dynamic graph visualization techniques (i.e., TEA and TET plot) together with the defined indices (i.e., *novelty*, *reoccurrence*, and *surprise* index) provide comprehensive summary of datasets characteristics. EdgeBank also provides a simple yet strong baseline that future dynamic link prediction methods can easily compare against. Additionally, our investigation on the impact of negative sampling in dynamic graphs leads to more robust evaluation setup for the dynamic link prediction task and facilitates methodological advancement in dynamic graph ML.

Since dynamic link prediction has many applications in different domains, such as recommendation systems, academic graphs, computational finance, etc., we expect this work to facilitate the development of applied methods in different domains as well. One potential negative impact is that future research may narrow down their study to these datasets. We aim to regularly update the datasets with the input from the community to prevent this issue. Additionally, improving link prediction can be associated with several potential negative use cases such as user profiling. While our work does not directly lead to such negative impacts, being aware of such impacts is important and appropriate precautions should be considered.

**Limitations:** We consider two main limitations for this work: First, in the current evaluation setup, there is a single point split for past and future links, which is the current common practice. It might be more relevant to consider alternative settings where temporal information plays a stronger role, from splitting in more time points to predict the exact time of an edge.

Second, we have only considered the *transductive* setting where all nodes are seen during training, since this is the only setup that we could easily check for memorization. The baseline and historic negative sampling strategy proposed here are only considered in the *transductive* setting.

In addition to these two main limitations, we only considered the dynamic link prediction task and leave the exploration of similar concepts in the related node classification task in dynamic graphs as future work.

**Acknowledgements:** This research is partially funded by the Canada CIFAR AI Chairs Program. The third author receives funding from IVADO. We thank Razieh Shirzadkhani for the help with cleaning and processing the Contact network.

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
