# OpenReview forum: "Towards Better Evaluation for Dynamic Link Prediction"
_NeurIPS.cc/2022/Track/Datasets_and_Benchmarks — NeurIPS 2022 Datasets and Benchmarks _

### Official Review · Reviewer_jFGW · 2022-07-25
**Review of "Towards Better Evaluation for Dynamic Link Prediction"**

**Rating:** 6
**Confidence:** 3
**Correctness:** Yes
**Clarity:** Yes

**Strengths:**

[+] 5 SOTA models are compared on 12 datasets in 3 different negative sampling settings.

[+] TEA & TET plots, EdgeBank and random NS show the limitations of current evaluation of dynamic link prediction.

[+] 5 new datasets are introduced.

**Weaknesses:**

[-]As this paper is a benchmark, it is better to include more DGNNs, like HTGN[1],EvolveGCN[2],DySAT[3],etc.

[-]As the paper adopts transductive setting, i.e. all nodes can be seen during training, static GNNs seem also able to tackle the tasks. It's better to include several representative static GNNs(like GIN[4],GraphSage[5],etc) as baselines to show whether they are beated by DGNNs in different NS settings.

[-]It is better to include the training and inference time for each DGNN.

[-]It is better to modularize the codes for convenience of interested researchers.

[1]Yang M, Zhou M, Kalander M, et al. Discrete-time temporal network embedding via implicit hierarchical learning in hyperbolic space[C]//Proceedings of the 27th ACM SIGKDD Conference on Knowledge Discovery & Data Mining. 2021: 1975-1985.

[2]Pareja A, Domeniconi G, Chen J, et al. Evolvegcn: Evolving graph convolutional networks for dynamic graphs[C]//Proceedings of the AAAI Conference on Artificial Intelligence. 2020, 34(04): 5363-5370.

[3]Sankar A, Wu Y, Gou L, et al. Dysat: Deep neural representation learning on dynamic graphs via self-attention networks[C]//Proceedings of the 13th international conference on web search and data mining. 2020: 519-527.

[4]Xu K, Hu W, Leskovec J, et al. How powerful are graph neural networks?[J]. arXiv preprint arXiv:1810.00826, 2018.

[5]Hamilton W, Ying Z, Leskovec J. Inductive representation learning on large graphs[J]. Advances in neural information processing systems, 2017, 30.

**Additional Feedback:**

1. Why use MLP instead of hadamard product operator[6] as downstream classifier?

[6]Grover A, Leskovec J. node2vec: Scalable feature learning for networks[C]//Proceedings of the 22nd ACM SIGKDD international conference on Knowledge discovery and data mining. 2016: 855-864.

**Documentation:**

Yes

**Relation To Prior Work:**

Yes

**Summary And Contributions:**

This paper aims at tackling the evaluation problem of dynamic link prediction. The authors propose two visualization techniques(TEA & TET plots) to illustrate the recurring patterns of edges over time, showing the difficulty of the datasets. They next show limitations of the current random negative sampling(NS) strategy and introduce two harder NS strategies, i.e. historical and inductive NS, to better test the generalization of DGNN models. A simple yet competitive memorization-based baseline EdgeBank is further proposed. 5 SOTA models are compared on 12 datasets in 3 different NS settings. It's found that performance and ranking of different models vary significantly.

---

> ### Author Response · Authors · 2022-08-23
> **Authors’ Response to Reviewer 5 (Reviewer jFGW)**
>
> We thank the reviewer for their time and interest in our work. We appreciate the reviewer’s detailed instructions on how to improve our work and we address the reviewer’s questions below:
>
> **Q1** As this paper is a benchmark, it is better to include more DGNNs, like HTGN[1], EvolveGCN[2], DySAT[3],etc.
>
> **A1** We thank the reviewer for the suggestion. The baselines suggested are categorized as “discrete-time” dynamic graph representation learning, while in this work we focus on “continuous-time” learning methods. However, we think it is valuable to add these baselines into our repository and bridge the gap between these methods, as they can be applicable once a given temporal network is aggregated into snapshots. We are exploring the inclusion of discrete-time methods, however this is not as straightforward as one might think. For datasets which have finer grain time information, this results in too many snapshots and the methods designed for discrete-time fail to run as they are designed based on datasets with small number of snapshots (e.g., about 20). We are looking into how to balance these and have a script that enables changing the settings from continuous to discrete, but this requires more time than allowed by this rebuttal period. We are planning to add these in future to the repository when ready.
>
> ---
>
> **Q2** As the paper adopts transductive setting, i.e. all nodes can be seen during training, static GNNs seem also able to tackle the tasks. It's better to include several representative static GNNs (like GIN[4],GraphSage[5],etc) as baselines to show whether they are beaten by DGNNs in different NS settings.
>
> **A2**  In order to apply a static graph method, the dynamic graph needs to be collapsed into a static representation. In this process, it is possible that much temporal information is lost. Therefore, we expect static graph methods to perform worse than DGNN methods. As a future direction, we believe it is possible to add these baselines. We thank the reviewer for this suggestion.
>
> ---
>
> **Q3** It is better to include the training and inference time for each DGNN.
>
> **A3** When running the experiments, we used different hardware for different models, e.g. we used more CPU cores for CAWN as it needs more CPU computation. Therefore we do not currently have accurate runtimes across the same hardware setup, and unfortunately due to computation time constraints we are not able to rerun all the experiments in time for this discussion. However, we agree with the reviewer that this is useful information and we will work to add it in future work.
>
> ---
>
> **Q4** It is better to modularize the codes for convenience of interested researchers.
>
> **A4** We plan to update our code base to be more accessible as part of our maintenance plan explained in Appendix E. In addition, we aim to host a dedicated website for the project with appropriate documentation on running the code and loading the datasets.
>
> ---
>
> **Q5** Why use MLP instead of hadamard product operator[6] as downstream classifier?
>
> **A5** Considering the conventional setting, we chose MLP because most of the methods that we examine use MLP as part of their default implementation. We would like to thank the reviewer for the suggestion. We consider this as an interesting future experiment to run.

---

### Official Review · Reviewer_Gob7 · 2022-07-26
**Review of "Towards Better Evaluation for Dynamic Link Prediction"**

**Rating:** 6
**Confidence:** 3
**Correctness:** The technical aspects of the work see…

**Strengths:**

- Although the novel baseline method is fairly simple (which is not a bad thing) it achieves outstanding results, even outperforming complex models on specific settings and datasets.
- The proposed novel negative edge sampling technique allows the differentiation of approaches based on specific objectives.
- New datasets introduced in the paper are a good contribution to diversity (different domains, different topological structures, different temporal edge evolution patterns).


**Weaknesses:**

- The introduction is confusing
  - Maybe include in the introduction for people not deep into dynamic link prediction: What are negative edges and how are they used?
  - The term "easy negative edges" is used as it would be commonly known terminology, while afaik it is not. Please explain it shortly. What are easy negative edges? (From the context it is more obvious that easy refers to negative edges that are not very similar to the true edge and thus evaluation performance is higher, however, I would advise stating this explicitly)
  - Either move Fig 1. to Results
    - Fig 1. confused and surprised me a little at this point in the paper, I would move it rather to the results or remove it completely (as it is already contained in Fig 5)
  - or Rework the caption of Fig 1.
    - Caption should describe what can be seen in the figure
    - "The ranking of different methods changes in the proposed negative sampling settings, which eliminate easy negatives." -> This sentence is confusing, please rework it. Make it clear what eliminates easy negatives and, as previously said, what easy negatives are.
    - Make it more clear that Historical and Inductive are your novel NS strategies.
  - What dataset was used for Fig 1.?
- Fig 3. is confusing
  - Fig 3. needs better description in text/caption, right now it is very confusing what can be seen in the plots. Here are some thoughts:
    - The X-Axis needs a less confusing label: With the current label "Percentage of observed edges" one would wrongly interpret the plot "(j) US Legis" at Timestamp 12 as that there are about 20% of observed edges in this timestamp which are orange? And then why is there another datapoint in Timestamp 12 that says about 27% of observed edges are orange? I think the X-Axis could be labeled in a more intuitive way.
    - Furthermore, is my assumption correct that a datapoint at a certain location X, represents the same edge at a different timestamp? This should be made more clear in the caption/text.


**Additional Feedback:**

N/A

**Clarity:**

The paper is well written and easy to read, however
- It irritates a little that reoccurrence and recurrence are used interchangeably throughout the text, maybe make that uniform where it makes sense.
- Typo L234: reocurring -> reoccurring

**Documentation:**

Dataset details are documented and a GitHub repository is provided.

**Ethics:**

There is no known ethical issue.

**Relation To Prior Work:**

No prior work reported.

**Summary And Contributions:**

In this work, the authors point out problems with the evaluation of dynamic link prediction approaches. As approaches often achieve near-perfect results, the authors argue that a) datasets are too simple and b) current sampling techniques generate negative edges that lead to less differentiation of approaches. The authors offer novel tools for better evaluation of dynamic link prediction and advise others to use them: a ) They introduce new more robust negative sampling techniques, b) strong baseline method for comparing approaches against, c) 5 novel dynamic graph datasets and two new techniques for generating datasets analysis plots.

---

> ### Author Response · Authors · 2022-08-23
> **Authors’ Response to Reviewer 4 (Reviewer Gob7)**
>
> We thank the reviewer for their time and interest in our work. We have used your feedback to improve the paper. We address your questions below:
>
> **Q1** Maybe include in the introduction for people not deep into dynamic link prediction: What are negative edges and how are they used?
>
> **A1** We thank the reviewer for this point. We have added explanation for negative edges in Section 1: Introduction.
>
> ---
>
> **Q2** The term "easy negative edges" is used as it would be commonly known terminology, while afaik it is not. Please explain it shortly. What are easy negative edges? (From the context it is more obvious that easy refers to negative edges that are not very similar to the true edge and thus evaluation performance is higher, however, I would advise stating this explicitly)
>
> **A2** We agree with the reviewer and have revised the paper to explain explicitly. Easy negative edges often refer to those edges that can be easily predicted by a simple baseline, such as our proposed EdgeBank. More specifically, easy negative edges in our context denote those edges which are never observed during training and they have never appeared in the memory component of a DGNN nor stored in EdgeBank memory. Therefore, it is unlikely that these edges will appear during the test phase; thus, for a given method, these are considered as easy to predict.
>
> ---
>
> **Q3** General write up suggestions. a). Either move Fig 1. to Results or or Rework the caption of Fig 1.  b). Make it more clear that Historical and Inductive are your novel NS strategies.
>
> **A3** We have updated the caption of Fig1. to avoid confusion. We now specify that our proposed negative sampling strategy contains more difficult negative edges.
>
> ---
>
> **Q4** What dataset was used for Fig 1.?
>
> **A4** We have illustrated Fig. 1 as an empirical motivation for two of our contribution:
> It shows that negative sampling is important in evaluation of different methods and different negative sampling strategies result in different ranking of the methods.
> It shows the considerably high performance of a simple memorization approach, like EdgeBank, especially for the standard negative sampling setting.
> The results presented in Fig. 1 demonstrate the average performance of each baseline method over all available datasets introduced in Table 1. We have updated Fig. 1 caption to include the missing information.
>
> ---
>
> **Q5**  Inconsistent usage of “reoccurrence” vs. “recurrence”
>
> **A5** Thank you for pointing this out, we have updated the paper to be consistent.
>
> ---
>
> **Q6** No prior work reported.
>
> **A6** We would like to point the reviewer to Section 2 of our submission which extensively reviews the literature on dynamic link prediction. We would also be happy to include any missing related work that the reviewer considers missing and it would be important to add.

---

### Official Review · Reviewer_yWmH · 2022-07-27
**Strong baseline and 5 new datasets**

**Rating:** 6
**Confidence:** 5
**Correctness:** The evaluation methods and experiment…
**Clarity:** The paper is well written and easy to…

**Strengths:**

1. The proposed visualization techniques, TEA and TET plots, and the corresponding indices are useful to quickly examine datasets characteristics.
2. New datasets and negative sampling strategies can facilitate the development of novel models.
3. The proposed EdgeBank will be a strong and widely-used baseline.

**Weaknesses:**

1. Only the transductive setting is considered in the paper. Inductive temporal link prediction is also an important setting to test the model performance on new nodes.
2. Though 5 new datasets are proposed, their time granularity is too coarse. This makes the datasets more like discrete-time dynamic networks (sequences of snapshots).

**Additional Feedback:**

1. (More on weaknesses #2) Most existing methods are not optimized for massive concurrent events [1]. For example, Jodie always processes events sequentially. Extra results on how the degree of concurrence of events will affect the models will be interesting.
2. Current methods have achieved near-perfect performance due to the weak negative sampling-based evaluation. Is it possible to show scores of MRR or Recall@k without sampling just like Jodie did?

[1] Zhang et al., CoPE: Modeling Continuous Propagation and Evolution on Interaction Graph. CIKM'21.

**Documentation:**

The comparing methods, evaluation pipelines, and visualization tools are accessible on GitHub with the detailed README.  There seems to be sufficient detail to support reproducibility.

**Relation To Prior Work:**

The authors clearly discuss how this work differs from previous papers in Related Work section.

**Summary And Contributions:**

The authors focus on representation learning and dynamic link prediction on continuous-time dynamic networks. The authors review the current evaluation procedures and datasets, summarize three main limitations, and then propose the corresponding solutions. 1. Most existing datasets are limited in the social interaction domain. So the authors provide 5 new datasets ranging from politics, economics, and transportation domains, and develop visualization techniques to examine different properties of datasets. 2. Current methods for dynamic link prediction have achieved near perfect performance due to the weak negative sampling-based evaluation. The authors thus propose 2 new sampling strategies. 3. Considering the re-occurring patterns in datasets, the authors propose a novel non-parameterized and memorization-based baseline EdgeBank.

---

> ### Author Response · Authors · 2022-08-23
> **Authors’ Response to Reviewer 3 (Reviewer yWmH)**
>
> We thank the reviewer for their time and interest in our work. We appreciate the reviewer’s detailed instructions on how to improve our work. We address reviewer’s questions below:
>
> **Q1** Only the transductive setting is considered in the paper. Inductive temporal link prediction is also an important setting to test the model performance on new nodes.
>
> **A1** In this work, we focus on the transductive setting which is applicable for almost all existing methods as well as our proposed baseline EdgeBank. A potentially different set of tools and protocols might be needed to properly evaluate the inductive setting. We thank the reviewer for pointing out the inductive temporal link prediction setting and we consider it as an important future direction.
>
> ---
>
> **Q2** Though 5 new datasets are proposed, their time granularity is too coarse. This makes the datasets more like discrete-time dynamic networks (sequences of snapshots). Most existing methods are not optimized for massive concurrent events [1]. For example, Jodie always processes events sequentially. Extra results on how the degree of concurrence of events will affect the models will be interesting.
>
> **A2** Among the newly proposed datasets, some datasets’ time granularity is restricted by their domain. For example the US Legis. network and Can. Parl. network are political networks which only publish data on an annual basis. We believe that the difference in time granularity is one of the novel aspects of our proposed datasets which will add to the benchmark dataset diversity. In addition, we now contribute another new dataset, Contact, a physical proximity network, has finer time-granularity (its time granularity is 5 minutes), and includes considerably more timestamps. The details of the Contact dataset are now given in Table 1. and further explained in Section 3.1, with the explanation of the preprocessing procedure provided in Appendix A.2.
> As the reviewer mentioned, methods, such as JODIE, excel in a sequential setting and the degree of concurrence of events might impact their performance. Understanding the impact of time granularity on the performance and/or computation/memory consumption of individual methods is also an interesting future direction.
>
> ---
>
> **Q3** Current methods have achieved near-perfect performance due to the weak negative sampling-based evaluation. Is it possible to show scores of MRR or Recall@k without sampling just like Jodie did?
>
> **A3** Computation of these metrics requires either going through every possible edge or directly predicting the most likely edges. The former is not computationally feasible for most datasets due to the large number of possible edges at potentially many timestamps. The latter does not suit the setting where a method should predict the existence of an edge between a pair of nodes based on their embeddings, which is the conventional setting in the dynamic link prediction task in many different applications. Thus, although MRR or Recall@k is a good suggestion that can be insightful in specific situations, in this work, we choose to focus on the more general setting where both positive and negative edges are tested.

---

### Official Review · Reviewer_3Auu · 2022-07-27
**Review - Interesting Benchmark for an Important Problem**

**Rating:** 7
**Confidence:** 4

**Strengths:**

* Dynamic Link Prediction is an important problem, and the paper offers very clear answers with elegant baseline approaches, visualization techniques, and benchmarks.
* the authors have provided detailed analysis and experiments to illustrate the problems of current benchmarks, as well as the superiority of the new approach + baseline.
* the ideas of "Easy Negative Edges" and "Memorization Works Well" are very fresh & insightful to me.

**Weaknesses:**

* all five proposed datasets seem to have significantly fewer unique timestamps than existing ones & are sampled in longer frequencies, granted that the nature of some of these datasets (Politics) determines such outcome.
* the core contributions of the paper seem to be split into three sub-points that are slightly incoherent - thought individually interesting, the main text seems to also split evenly introducing each of them, therefore losing a bit of focus.

**Additional Feedback:**

### Questions

* does the "near-perfect performance of dynamic link prediction methods" refer to the "Standard" setting in figure 1 and 5?
* is the new benchmark designed to be evaluated in a suite? If so, the additional five datasets don't seem to really solve the limited domain diversity problem as it only expands to politics, economics, and transportation. Domains such as "biological networks such as protein interaction" remain unaddressed.
* does LIWC in the appendix refer to https://www.liwc.app/help/howitworks?
* relating to the subject of backtesting in time series forecasting, is it possible to incrementally build $E_{\text{train}}$ and $E_{\text{test}}$ based on a series of time splits? Would this benchmark still be useful?

**Clarity:**

Apart from the focus point in weakness, this paper is very well written and easy to understand. The length of the main text and supplemental materials are also appropriate.

**Correctness:**

The construction of dataset, experiment design, and evaluation methods appear to be correct.

**Documentation:**

The dataset and benchmark are adequately documented, with a few questions:
* intended uses are included
* license is included
* data is easily accessible (via Google Drive), though it is unclear what the maintenance plan is
* additionally, it is unclear how the new datasets are created from [5, 11, 21, 32]
* there are sufficient details to reproduce the experiments.


**Ethics:**

There seem to be no ethical concerns for the datasets, and the authors seem to be appropriately addressing the negative impacts of dynamic link prediction in the appendix.

**Relation To Prior Work:**

This work is sufficiently different from previous contributions and well discussed.

**Summary And Contributions:**

The authors propose a more robust procedure for evaluating dynamic link prediction on graphs. By noticing the disconnect between existing benchmarks and real-world uses (easy negative edges unchecked and the strength of simple memorization), the authors propose three key contributions as part of a larger benchmark suite:
* Two novel sampling strategies on graphs: historical Negative Sampling (NS) and inductive NS, which provides more insight into the effects of such on dynamic link prediction benchmarks
* Two variants of EdgeBank, a novel but simple, memorization-based approach that beats many state-of-the-art methods on existing benchmarks, thus illustrating their weaknesses
* Finally, five new datasets that expand the domain of current benchmarks and two visualization tools (Temporal Edge Appearance Plot and Temporal Edge Traffic Plot) to help better understand the benchmark.

---

> ### Author Response · Authors · 2022-08-23
> **Authors’ Response to Reviewer 2 (Reviewer 3Auu): Part 1**
>
> We thank the reviewer for their time and interest in our work. We appreciate the reviewer’s detailed instructions on how to improve our work and we address reviewer’s questions below:
>
> **Q1** Does the "near-perfect performance of dynamic link prediction methods" refer to the "Standard" setting in figure 1 and 5?
>
> **A1** Yes, this is referring to the Random Negative Sampling setting or the standard setting used in the existing literature. We discuss the Random Negative Sampling setting in detail in Section 5.
>
> ---
>
> **Q2** Is the new benchmark designed to be evaluated in a suite? If so, the additional five datasets don't seem to really solve the limited domain diversity problem as it only expands to politics, economics, and transportation. Domains such as "biological networks such as protein interaction" remain unaddressed.
>
> **A2** We concur with the reviewer that biological networks are a worthwhile future domain. We believe this work is the first step towards creating a comprehensive benchmark suite which can enable and motivate further progress in the field of dynamic graph learning. The six novel datasets provide a starting point for increasing domain diversity. We plan to include additional datasets in novel domains as part of our future maintenance plan.
>
> ---
>
> **Q3** Does LIWC in the appendix refer to https://www.liwc.app/help/howitworks?
>
> **A3** By LIWC, we mainly refer to the following reference which is referred to by JODIE [18] (JODIE is the first paper that proposes most of the previously existing dynamic graph datasets):
> * Pennebaker, J.W., Francis, M.E. and Booth, R.J., 2001. Linguistic inquiry and word count: LIWC 2001. Mahway: Lawrence Erlbaum Associates, 71(2001), p.2001.
>
> ---
>
> **Q4** Relating to the subject of backtesting in time series forecasting, is it possible to incrementally build Etrain and Etest based on a series of time splits? Would this benchmark still be useful?
>
> **A4** In our work, we consider the standard split where all timestamps are chronologically split into three disjoint chunks as training, validation and test sets. While this is the conventional setting employed by most of the baselines for dynamic link prediction, it is also possible to construct the split differently such as incrementally building Etrain and Etest as the reviewer suggested. One example would be to incrementally include the newest timestep into the training set and ask the model to predict the future time step given a growing training set. It can be quite practical for real-world deployment where the task is to predict the future on a short-term, e.g., daily or hourly, basis (such as traffic forecasting). Discussion of this setting would be promising future work and we thank the reviewer for pointing out this direction.

---

> > ### Author Response · Authors · 2022-08-23
> > **Authors’ Response to Reviewer 2 (Reviewer 3Auu): Part 2**
> >
> > **Q5** All five proposed datasets seem to have significantly fewer unique timestamps than existing ones & are sampled in longer frequencies, granted that the nature of some of these datasets (Politics) determines such outcome.
> >
> > **A5** As mentioned by the reviewer, the number of timestamps in a dataset such as the US Legis. network and the Can. Parl. network are due to the nature of the data. Although they are not small datasets in terms of the number of edges or unique edges, they have coarser timesteps. We believe that this difference can also contribute to the diversity of benchmark datasets and facilitate testing methods in both coarse- and fine-grained time granularity temporal networks. Moreover, we now contribute one additional dataset, Contact, which has considerably more timestamps compared to other datasets that we proposed. The detailed statistics of this dataset are provided in Table 1 in and its preparation process is explained in Appendix A.2 of the paper.
> >
> > ---
> >
> > **Q6** The core contributions of the paper seem to be split into three sub-points that are slightly incoherent - thought individually interesting, the main text seems to also split evenly introducing each of them, therefore losing a bit of focus.
> >
> > **A6** As mentioned in the abstract and introduction, the high level theme of the paper is re-evaluation of the current testing protocols for the dynamic link prediction task. We demonstrate that the current evaluation protocol, which consists of the three sub-points mentioned by the reviewer, should be improved in terms of each of the different parts – i.e. dataset diversity, negative sampling, and baselines – in order to improve the robustness of the evaluation of the dynamic link prediction task. And we propose ways to make such improvements, e.g. new and more diverse datasets, new informative negative sampling techniques, new visualization techniques, and a simple strong baseline. In the revised manuscript, we aim to make these three points more coherent as suggested.
> >
> > ---
> >
> >
> > **Q7** data is easily accessible (via Google Drive), though it is unclear what the maintenance plan is
> >
> > **A7** To host datasets in a more permanent fashion, we now host them on Zenodo. We also added a detailed maintenance plan in Appendix E. Key points of the maintenance plan are the inclusion of additional datasets, methods and documentation.
> >
> > ---
> >
> > **Q8** Additionally, it is unclear how the new datasets are created from [5, 11, 21, 32]
> >
> > **A8** We added details for the collection process of each of the proposed datasets, how each dataset was converted to a dynamic graph format and information regarding its real-world interpretation in Appendix A.2.

---

> > > ### Comment · Reviewer_3Auu · 2022-08-23
> > > **Response**
> > >
> > > Thank you for your response. I've updated my review accordingly.

---

### Official Review · Reviewer_WjMG · 2022-07-28

**Rating:** 7
**Confidence:** 4
**Clarity:** Yes, the paper is well written and un…

**Strengths:**

The paper presents some new insights for evaluating link prediction on dynamic graph networks and exploration of properties of dynamic network datasets. The authors introduced new datasets concerning new domains (Politics, Transport, Economics), which may lead to methods that are more domain-concerned. The authors also provide detailed results in an appendix and discuss different performance metrics for link prediction.

**Weaknesses:**

* There are some concerns regarding the influence of the node/edges attributes, which are further discussed
* There are some concerns about the availability of datasets, which are further described
* The authors state that "if the number of available historical edges is insufficient to match the number of positive edges, the remaining negative edges are sampled by the random NS strategy." I would strongly recommend adding information about how many negative edges were sampled according to a random strategy for all evaluated datasets. These should be done in both Historical NS and Inductive NS

**Additional Feedback:**

Authors are advised to:

- Change the strength of some statements
- Provides more details on datasets' edge/node attributes and how models utilize them.
- Consider the aspect of dataset attributes to the presented results
- Improve availability of datasets, remove or fix redundant files (like files consisting of zero tensors)

**Correctness:**

There are some concerns about the submission correctness:

* The authors state that ”existing implementations have no collision check between positive and negative edges.” That is not generally valid, as some methods have already considered collision checking. E.g., FILDNE sampling procedure https://gitlab.com/fildne/fildne/-/blob/master/dgem/tasks/link_prediction/dataset.py
* The authors use attributed datasets. However,  in Table 1. with the datasets overview, there is no information about the node and edge features, which should be updated.
* In Appendix A.1. there for all models (except TGN), the authors do not state whether node/edge features are utilized. Also, that aspect is not covered in the discussion of the results.


**Documentation:**

There are no details of data collection and data cleaning of the proposed datasets. Also, there are some issues with availability and maintenance:

* The maintenance plan for the proposed datasets is missing. All datasets are shared via google drive in a single zipped file. There can be concerns about the availability of that dataset in the future, as data may be longer available on google drive (link may be changed / data can be removed). The authors added that link on GitHub, but that does not improve availability.
* The authors provided “.npy” files for some datasets with edge or node features. After reading these files via NumPy loader, we can see that tensors consist only of zeros.
* There is a difference in naming between paper and shared datasets like the Flights dataset is named as “covid_20200301_20200630.csv”.

**Ethics:**

Authors are aware of the ethical concerns of published new datasets and the negative impacts of further development of the link prediction task

**Relation To Prior Work:**

Relation to prior work is discussed.

**Summary And Contributions:**

The paper proposes two new techniques for the negative sampling of dynamic link prediction and two new visualization techniques and introduces five new dynamic graphs datasets from under-evaluated domains like politics, transport, and economics.

The paper is clear and understandable. The further examination resulted in several concerns. Addressing these concerns during the rebuttal may warrant an increased score.

---

> ### Author Response · Authors · 2022-08-23
> **Authors' Response to Reviewer 1 (Reviewer WjMG): Part 1**
>
> We thank the reviewer for their time and interest in our work. We appreciate the reviewer’s detailed instructions on how to improve our work and we address the reviewer’s questions below:
>
> **Q1** Change the strength of some statements
>
> **A1** We thank the reviewer for pointing out the reference for FILDNE. We changed the statement to be more general: “most existing implementations have no collision check between positive and negative edges.”
>
> ---
>
> **Q2** Provides more details on datasets' edge/node attributes and how models utilize them. In Table 1. of the datasets overview, there is no information about the node and edge features, which should be updated.
>
> **A2** We added more discussion in Section 3, including information about the edge features for each dataset. However, none of the datasets have node attributes, and when required by a baseline in our experiments, a vector of all zero is passed instead. We have clarified this in Appendix A.1.
>
> ---
>
> **Q3** Consider the aspect of dataset attributes to the presented results
>
> **A3** As all methods utilize the available edge weights, they utilize the same information for training and testing.
>
> ---
>
> **Q4** There are no details of data collection and data cleaning of the proposed datasets.
>
> **A4** For all proposed datasets, we added details for the collection process of the dataset, how each dataset was converted to a dynamic graph format and information regarding its real-world interpretation in Appendix A.2.
>
> ---
>
> **Q5** remove or fix redundant files (like files consisting of zero tensors). The authors provided “.npy” files for some datasets with edge or node features. After reading these files via NumPy loader, we can see that tensors consist only of zeros.
>
> **A5** According to the reviewer’s comment, we modified the dataset files and uploaded a cleaned version on Zenodo. Additionally, we would like to provide more clarification here. To prepare the network datasets, we closely followed the state-of-the-art baseline methods (i.e., JODIE [18], TGAT [42], TGN [28], and CAWN [41]). For all the datasets, the original edge-list is saved as <network>.csv. This file contains the timestamped edge-list of the networks in addition to the edge attributes if available.
> However, the baseline methods employ pre-processed versions of the datasets (the pre-processing can be done by using the utility script provided in the project repository). Specifically, in the preprocessing step, we generate three files for each dataset as follows:
>
> * <ml_network>.csv: contains the timestamped edge-list.
>
> * <ml_network>.npy: contains the edge features in dense ‘npy’ format.
>
> * <ml_network_node>.npy: contains the node features in dense ‘npy’ format.
>
> It is noteworthy that when the node or edge features are absent, a vector of zeros is used as the initial node/edge features.
> As none of these datasets have any node features, the <ml_network_node>.npy files contain only zeros.
> Although the node/edge feature files may only contain vectors of zeros, we decided to upload these pre-processed files as a ready-to-use version of the datasets for accessibility, especially considering that the file sizes are reasonable. It should be noted that the original datasets are also accessible if needed. We have provided explanations in the shared folder on https://zenodo.org/record/7008205#.YwJzpHbMIuW as well.

---

> > ### Author Response · Authors · 2022-08-23
> > **Authors' Response to Reviewer 1 (Reviewer WjMG): Part 2**
> >
> > **Q6** The maintenance plan for the proposed datasets is missing. All datasets are shared via google drive in a single zipped file.
> >
> > **A6** To host the dataset externally and to increase accessibly, we now host the datasets on Zenodo. The access link is https://zenodo.org/record/7008205#.Yv_a_3bMJPZ. The maintenance plan is added in Appendix E. For the maintenance plan, we aim to add more datasets, methods and documentation in the future.
> >
> > ---
> >
> > **Q7** The authors state that "if the number of available historical edges is insufficient to match the number of positive edges, the remaining negative edges are sampled by the random NS strategy." I would strongly recommend adding information about how many negative edges were sampled according to a random strategy for all evaluated  datasets. These should be done in both Historical NS and Inductive NS.
> >
> > **A7** We have added a discussion in Appendix B.2 and detailed statistics are reported in Table 12.  In summary, in the historical negative sampling setting, all the datasets have enough historical edges, thus no random sampling is done. In the inductive negative sampling setting, based on the dataset, we have zero to significant number of randomly sampled edges. An extreme example is Social Evo. dataset. As it is illustrated in Social Evo.’s TET plot in Fig. 3.e, this dataset does not have enough inductive edges, therefore the majority of the negative edges during the test phase are randomly selected. The detailed statistics for all datasets in both settings can be found in Table 12.
> >
> > ---
> >
> > **Q8**  There is a difference in naming between paper and shared datasets like the Flights dataset is named as “covid_20200301_20200630.csv”
> >
> > **A8** Thank you for pointing this out. We revised the datasets and made sure that there is consistency between the datasets names and the shared files.

---

> > > ### Comment · Reviewer_WjMG · 2022-08-29
> > > **Reviewer Response**
> > >
> > > Thanks for your detailed response and the significant improvements over the first version. It is worth appreciating significant changes regarding datasets storage and documentation, providing additional information in the supplementary materials, or adding a new dataset to the benchmark. The final score was updated accordingly.

---

### Author Response · Authors · 2022-08-23
**Summary of Changes and Revised Manuscript**

We would like to thank all reviewers for the detailed reviews as well as many constructive and positive comments. We are encouraged that reviewers find our suggested improvements to the current dynamic link prediction evaluation as valuable contributions to the dynamic graph learning community. We have followed the reviewers’ feedback to improve the paper, and revised a new version of the paper to reflect the changes (a standard version with all text in black and a marked version where changes are colored blue in the supplementary material). Based on the reviews, we have improved the paper from several aspects that are summarized as follows:

* To ensure long term access to the datasets, we now host them on Zenodo (Digital Library for Open Data by CERN). The access link is https://zenodo.org/record/7008205#.YwQfPnbMJPa.

* We describe our dataset maintenance plan in detail in Appendix E. In short, we plan to regularly update the project repository with more datasets, methods and documentation for a more robust and accessible benchmark.

* We expanded our benchmark by adding another novel temporal Contact network dataset (see Section 3.1), and added experiments on this dataset for the three negative sampling settings.

* We polished the writeup to include additional details, such as an explanation of data collection and processing for our proposed datasets (see Appendix A.2), and edge/node attribute information in Section 3.

Again, we appreciate all reviewers for their valuable suggestions which help us to further improve our work. We expect our updated version to address the main comments of reviewers.

---

> ### Author Response · Authors · 2022-08-29
> **Following up on the discussion**
>
> We would like to thank all reviewers for their time and interest in our submission and we are glad to address any pending concerns.

---

### Meta-Review · Area_Chair_Ezoe · 2022-09-13

**Recommendation:** Accept
**Confidence:** 4

**Metareview:**

This paper presents multiple contributions towards understanding, evaluating, and developing better methods for dynamic link prediction. This includes:
 * New sampling strategies that better match real-world applications
 * A simple but surprisingly strong baseline, which already beats several more sophisticated methods
 * A new collection of 5 datasets for dynamic link prediction from different domains

Collectively, this is a significant advance supporting further research in dynamic link prediction. All reviewers were positive in their assessments of the paper, especially after the authors improved their paper in response to reviewer comments.

I recommend this for a spotlight because I think it could have significant impact within its area.

---

### Decision · Program_Chairs · 2022-09-16

Accept